# Advances in Pharmacological Approaches for Managing Hypercholesterolemia: A Comprehensive Overview of Novel Treatments

**DOI:** 10.3390/biomedicines12020432

**Published:** 2024-02-14

**Authors:** Andrea Mormone, Giovanni Tortorella, Francesca Esposito, Alfredo Caturano, Aldo Marrone, Domenico Cozzolino, Raffaele Galiero, Raffaele Marfella, Ferdinando Carlo Sasso, Luca Rinaldi

**Affiliations:** 1Department of Advanced Medical and Surgical Sciences, “Luigi Vanvitelli” University of Campania, 80131 Naples, Italy; andrea.mormone@studenti.unicampania.it (A.M.); giovanni.tortorella@studenti.unicampania.it (G.T.); francesca.esposito4@studenti.unicampania.it (F.E.); alfredo.caturano@unicampania.it (A.C.); aldo.marrone@unicampania.it (A.M.); domenico.cozzolino@unicampania.it (D.C.); raffaele.galiero@unicampania.it (R.G.); raffaele.marfella@unicampania.it (R.M.); ferdinandocarlo.sasso@unicampania.it (F.C.S.); 2Department of Experimental Medicine, “Luigi Vanvitelli” University of Campania, 80131 Naples, Italy; 3Department of Medicine and Health Sciences “Vincenzo Tiberio”, Università degli Studi del Molise, 86100 Campobasso, Italy

**Keywords:** hypercholesterolemia, cardiovascular disease, atherosclerosis, statins, lipid-lowering drugs

## Abstract

Hypercholesterolemia plays a crucial role in the formation of lipid plaques, particularly with elevated low-density lipoprotein (LDL-C) levels, which are linked to increased risks of cardiovascular disease, cerebrovascular disease, and peripheral arterial disease. Controlling blood cholesterol values, specifically reducing LDL-C, is widely recognized as a key modifiable risk factor for decreasing the morbidity and mortality associated with cardiovascular diseases. Historically, statins, by inhibiting the enzyme β-hydroxy β-methylglutaryl-coenzyme A (HMG)-CoA reductase, have been among the most effective drugs. However, newer non-statin agents have since been introduced into hypercholesterolemia therapy, providing a viable alternative with a favorable cost–benefit ratio. This paper aims to delve into the latest therapies, shedding light on their mechanisms of action and therapeutic benefits.

## 1. Introduction

Atherosclerosis is a chronic inflammatory disease that affects the endothelium of large and medium arteries, resulting in the formation of lipid plaques through interactions with other risk factors. Well-recognized contributors to atherosclerosis include hypercholesterolemia, arterial hypertension, diabetes mellitus, obesity, cigarette smoking, and numerous others [1,2,3,4,5]. Our research group conducted a clinical trial, wherein it was reported that the number of risk factors not at target is significantly associated with an increased cardiovascular risk. This investigation emphasizes the paramount significance of addressing a comprehensive array of risk factors to mitigate cardiovascular events [4]. Lipids constitute a pivotal modifiable risk factor for atherosclerotic disease (ASD). The early stages of atherogenesis involve the accumulation and retention of apolipoprotein B (apoB)-containing lipoproteins, particularly low-density lipoproteins (LDL), within arterial intima [6]. Elevated low-density lipoprotein (LDL-C) levels are associated with an increased risk of cardiovascular disease (CAD), cerebrovascular disease, and peripheral arterial disease (PAD) [7,8,9]. Efficient control of blood cholesterol, especially the reduction in LDL-C, stands as a reliable and achievable modifiable risk factor, significantly reducing the morbidity and mortality associated with cardiovascular disease (CVD) [10,11]. Evidence underscores that the extent and duration of LDL exposure determine the risk of ASD and its complications, underlining the urgency of identifying and treating elevated LDL-C levels [12].

Lovastatin, the first statin, was discovered in the 1970s, but it was only commercially available from 1987, gaining increasing notoriety since 1997 due to the demonstration, in the historic Scandinavian Simvastatin Survival Study, that statins can reduce the risk of cardiovascular disease (CVD) (HR 0.70, 95% CI 0.58, 0.85) [13]. Subsequently, various statin compounds with varying potency in reducing LDL-C levels have been marketed [14]. Statins reduce LDL-C levels by inhibiting the enzyme β-hydroxy β-methylglutaryl-coenzyme A (HMG)-CoA reductase in the cholesterol synthesis pathway, leading to an increased expression of hepatic LDL receptors and, therefore, to a greater absorption of LDL-C from the circulation, reducing its plasma levels [15,16]. Since the 1960s, pharmaceutical companies have started developing non-statin drugs with the sole aim of reducing cholesterol levels, including fibric acid derivatives (e.g., clofibrate, fenofibrate, gemfibrozil), niacins, and bile acid sequestrants/resins (e.g., cholestyramine, colesevelam). Fibrates, agonists of peroxisome proliferator-activated receptor alpha (PPAR-alpha), reduce blood triglyceride levels, increase high-density lipoprotein cholesterol (HDL-C) values, and minimally reduce LDL-C levels [17]. Niacins modulate lipolysis in adipose tissues but have limited tolerability and several side effects [18]. Bile acid resins bind bile acids in the intestine, limiting their reabsorption, eliminating the complex with the feces. Reduced tolerability and dosage formulations limit their use [19]. Another key non-statin therapy is ezetimibe, a well-tolerated drug with modest cardiovascular (CV) risk reduction [20,21]. Ezetimibe selectively inhibits cholesterol absorption by binding to the sterol transporter Niemann–Pick C1-Like 1 (NPC1L1; SLC65A2) in the small intestine, decreasing cholesterol release into the mesenteric veins and increasing its clearance in the blood [22].

Statins, supported by numerous randomized clinical trials, have been the first-line therapy in the primary and secondary prevention of ASD [23,24]. In 2015, the Food and Drug Administration (FDA) approved alirocumab and evolocumab, monoclonal antibodies against proprotein convertase subtilisin/kexin type 9 (PCSK9), representing the most advanced approach for PCSK9 inhibition. PCSK9 is a hepatic secretory protein that negatively regulates the LDL receptor (LDL-R) by binding to its extracellular domain, mediating its internalization and degradation. This process increases the circulating levels of LDL-C and prevents its absorption [25,26]. Monoclonal antibodies (e.g., evolocumab, bococizumab, alirocumab) work synergistically with statins, reducing LDL-C levels by approximately 60% and lowering cardiovascular event incidence [27,28,29].

In 2013, the primary objective of the American College of Cardiology (ACC) and American Heart Association (AHA) guidelines for hypercholesterolemia treatment shifted to identifying and managing ASD risk rather than treating lipid values alone [30]. Minimal guidance on non-statin therapies was provided, recommending their use in high-risk patients with an unsatisfactory statin response or statin intolerance [30]. In 2016, the ACC designated ezetimibe as the non-statin therapy of choice for clinically stable ASD patients not reaching their LDL-C target values [17]. PCSK9 inhibitors and bile acid sequestrants/resins are considered alternatives to ezetimibe, while niacin is not recommended due to a lack of evidence for its benefit [17].

The 2019 European Society of Cardiology (ESC) and European Atherosclerosis Society (EAS) guidelines updated their recommendations for primary prevention with statin therapy, modifying the 2016 guidelines [31,32]. The general therapeutic approach involves assessing total cardiovascular risk, establishing objectives based on risk, involving the patient in CV risk management decisions, and increasing statin dosage before exploring additional interventions to achieve the selected goal (e.g., ezetimibe or PCSK9 inhibitors) [31]. The purpose of this review is to outline all the available therapies for hypercholesterolemia and their impact on the prevention of overall cardiovascular risk.

### Data Collection

A literature review was conducted, updated on 30 December 2023, using Pub-Med/MEDLINE, Scopus, and Web of Science. We employed a combination of the following keywords: (1) “hypercholesterolemia” OR “low-density lipoprotein” AND “high-density lipoprotein cholesterol” OR “atherosclerotic cardiovascular disease” OR “Diabetes” OR “lipidic plaque” OR “Metabolic Syndrome” OR “Statins” AND “non-statin therapies.” Additionally, a manual search for additional publications, both in clinical and pre-clinical studies, was carried out to include any potentially missed through electronic searches. Duplicate records were excluded, along with articles that were not available in English or were deemed ineligible by automation tools (Figure 1).

## 2. Currently Available Drugs

### 2.1. Mipomersen

Mipomersen is a second-generation drug designed to reduce the levels of LDL-C by inhibiting the synthesis of apolipoprotein B-100 (apoB). The U.S. Food and Drug Administration has approved mipomersen as third-line therapy for patients with homozygous familial hypercholesterolemia (HoFH) [33,34]. It functions as an antisense oligonucleotide, binding complementarily to human apo-B 100 messenger RNA (mRNA) [35]. This binding triggers the recruitment of a specific catalytic enzyme (Rnase H1), leading to the degradation of the apo-B RNA complex [36]. By reducing the cytoplasmic apo-B 100 mRNA concentration, mipomersen plays a role in lowering the final production of LDL-C.

Administered subcutaneously once a week in a preformulated syringe with 200 mg/mL, mipomersen exhibits good absorption and distribution throughout the human body. It undergoes metabolism not by cytochrome P450 in the liver but by endonucleases in the tissues, with its metabolites ultimately eliminated in the urine. Common adverse events include injection-site reactions (erythema, pruritus, pain), and a flu-like syndrome is reported in about 30% of cases. Clinical trials have indicated the potential for hepatotoxicity; thus, constant monitoring of serum alanine aminotransferase (ALT), aspartate transaminase (AST), total bilirubin, and alkaline phosphatase is crucial during treatment [37,38]. Mipomersen should be discontinued if hepatic cytolysis index levels exceed three times the normal range in the serum or if liver toxicity becomes clinically significant [39]. Contraindications for mipomersen include hypersensitivity/allergic reactions and moderate or severe hepatic impairment (Child–Pugh B or C). In patients with HF and affected by documented atherosclerotic cardiovascular disease (ASCVD), mipomersen significantly reduced apolipoprotein B by 26.3%, total cholesterol by 19.4%, and lipoprotein(a) by 21.1% compared to the placebo (all *p* < 0.001). These reductions contribute to the improvement of atherosclerotic disease [40].

### 2.2. Lomitapide

Lomitapide is a novel drug designed to lower the cholesterol concentration in the blood. It is indicated for the treatment of patients with HoFH who have an inadequate response to PCSK9 inhibitors. Additionally, it is prescribed for patients with ASCVD whose LDL-C levels exceed 190 mg/dL and do not respond to statin treatment. It has proven effective in reducing cholesterol levels in patients with HoHF, showing reductions in LDL-C and apoB of up to 51% and 56%, respectively. However, there is still a lack of evidence regarding the overall improvement in survival in patients with ASCVD [41,42,43,44,45,46]. In contrast to the three main classes of cholesterol-lowering drugs (statins, ezetimibe, and bile acid sequestrants), lomitapide acts independently of the expression of LDLR. It achieves this by inhibiting the microsomal triglyceride transfer protein (MTP) in the endoplasmic reticulum lumen, effectively binding to and deactivating MTP. MTP is crucial in cholesterol synthesis as it transports lipid molecules (triglycerides, phospholipids, and cholesterol esters) to apoB-lipoproteins in the endoplasmic reticulum, facilitating the formation of very-low-density lipoprotein (VLDL), low-density lipoprotein (LDL), and chylomicrons [47].

Lomitapide is taken orally on an empty stomach, preferably at least 2 h after meals to mitigate the risk of gastrointestinal side effects. The initial dose is 5 mg per day, gradually increasing over 2–4 weeks to a maximum dose of 60 mg per day if well tolerated. The primary side effects include nausea, vomiting, diarrhea, abdominal pain, and potential liver toxicity. While lomitapide may elevate transaminases, clinically significant elevations in alkaline phosphatase, total bilirubin, or the international normalized ratio (INR) are rare [48,49]. Monitoring liver and renal function tests before initiating treatment and at least monthly during the first year is recommended. Contraindications include pregnancy, liver impairment (Child–Pugh B or C) or unknown increased levels of transaminases [50], and hypersensitivity to the active principle.

### 2.3. Inclisiran

Inclisiran, a novel drug approved for hypercholesterolemia treatment, acts by inhibiting PCSK9 protein translation in hepatocytes [51]. PCSK9, produced in the liver, is involved in the degradation process of LDL-R on the cell membrane. Inclisiran, a small interfering ribonucleic acid molecule (siRNA), interacts with the RNA-induced silencing complex (RISC), cleaving PCSK9 mRNA, preventing translation of the target protein [52]. The reduction in PCSK9 increases LDL-R availability on the membrane, leading to a greater uptake of circulating LDL and a reduction in serum LDL-C.

The regimen, based on the ORION-1 study’s results, involves administering a dose of 284 mg with one subcutaneous injection on day 1, another on day 90, and then one administration every 6 months [53]. The results from the ORION-1 study with a 300 mg dose showed an average reduction of 52.6% (48% to 71%) in LDL-C levels at day 180, with a mean reduction of 47.2% at day 240 after receiving the two doses. In these studies, a reduction in PCSK9 levels by an average of 69.1 ± 12.1% at day 180 and at day 240 of 40% was observed. A one-year follow-up of the ORION-1 participants showed an LDL-C reduction of 31.4% at 360 days, signifying a sustained but waning effect over time [54].

The inclisiran phase 3 trials, particularly ORION-9, evaluated the efficacy of inclisiran in patients with HoFH already treated with the maximum dose of statin and ezetimibe with a baseline level of LDL-C more than 100 mg/dL; it showed an average reduction of 47.9% (95% CI −53% to −42.73 *p* ˂ 0.001) compared to a placebo [55]. ORION-10 and -11 evaluated the efficacy of inclisiran in patients with ASCVD or ASCVD equivalent risk and LDL levels of more than 70 mg/dL; in this study, it was seen how inclisiran is able to lead to an average reduction of 52.3% (−52.3% (95% CI −55.7 to −48.8%; *p* < 0.001), thus proving effective in improving ASCVD [56].

Regarding the impact of inclisiran in cardiovascular outcomes trials, the results are still ongoing; however, a recently published pooled patient-level analysis of ORION-9, -10, and -11 showed that the occurrence episodes of major adverse cardiovascular events (MACE) (131 vs. 172 events; hazard ratio (HR) 0.75, 95% CI 0.60–0.94), fatal/non-fatal MI (33 vs. 41 events; HR 0.81, 95% CI 0.51–1.29), and fatal/non-fatal stroke (13 vs. 15 events; HR 0.80, 95% CI 0.39–1.67) were all lower in the group of patients treated with inclisiran [57]. This is indicated in the treatment of hypercholesterolemia in patients with HeFH or ASCVD already treated with the maximum dose of statins and ezetimibe that require further lowering of LDL-C [58]. Off-label, it could be useful also in patients with contraindication or intolerance to statin treatment.

Safety and side-effects were evaluated in all of these three clinical trials; the most common side effects are injection site reactions (5% in the inclisiran vs. 0.7% in the placebo group; risk ratio 7.54), bronchitis (4.3% for inclisiran vs. 2.7% placebo; risk ratio 1.55), hypertension (5.7% for inclisiran vs. 5.7% for placebo), arthralgia (5.0% for inclisiran vs. 4.0% for placebo), back pain (4.5% for inclisiran vs. 4.2% for placebo), urinary tract infection (4.4% for inclisiran vs. 3.6% for placebo), and an increase in serum creatine phosphokinase (2.3% in inclisiran vs. 3.2% in placebo) [59]. Longer-term information about safety and tolerability is not currently available; however, recently published data from the ORION-4 study showed good results [60]. The safety data of inclisiran in patients with CKD, severe hepatic dysfunctions, and pregnancy are not yet available; however, data suggest that it could be considered safe in patients with CKD [61].

### 2.4. Bempedoic Acid

The need for increasingly stringent targets in controlling LDL-C values for preventing cardiovascular risk, coupled with the not-very-strict adherence to statin treatment [62], has spurred growing interest in developing new oral drugs for cholesterol therapy. Bempedoic acid (formerly ETC1002), a long-chain tetramethyl-substituted keto diacid, is a molecule first studied in 2003 that belongs to the family of so-called “fraudulent fatty acids”, along with fibrates, ω-3 fatty acids, and pantethine [63]. This acid inhibits ATP citrate lyase (ACLY), a cytosolic enzyme involved in lipid and glycide metabolism and synthesis. Specifically, ACLY plays a role in the complex reaction transforming citrate into acetyl-coA, a fundamental precursor for HMG-coA synthesis, an essential substrate for cholesterol production [64]. Blocking this enzyme increases LDL-R on the membrane, resulting in reduced circulating LDL, lipid reduction, decreased hepatic steatosis, and increased weight loss [65,66].

Bempedoic acid exhibits good absorption after oral administration, with excellent gastrointestinal tolerability and bioavailability. It is strongly bound to plasma proteins, possesses a half-life of about 21 h, and is primarily metabolized by the liver through hepatic glucuronidation [67]. Metabolism by cytochrome P-450 is minimal, avoiding significant drug interactions and allowing administration in patients with mild–moderate renal impairment [68]. Currently available commercially in 180 mg tablets [69], it is also available in a fixed combination with 10 mg of ezetimibe [70]. Indications include treating adult patients with hypercholesterolemia (familial and non-familial) and mixed dyslipidemia in addition to statins if the target is unattainable with the maximum tolerated dose of statins and ezetimibe or when statins are not tolerated or contraindicated.

The tolerability and safety profile were addressed naïvely the CLEAR program, comprising four phase 3 trials: (1) CLEAR Tranquility (in statin-intolerant patients) [71]; (2) CLEAR Harmony (patients with LDL-C ≥ 70 mg/dL despite maximally tolerated statin therapy) [72]; (3) CLEAR Wisdom (patients with ASCVD, HeFH, or both, on optimal statin treatment) [73]; and (4) CLEAR Serenity (statin-intolerant patients with ASCVD and inadequately controlled LDL-C) [74]. Despite the excellent results of bempedoic acid in the CLEAR studies in terms of LDL-C reduction (18% in combination with statin and 24% when administered in patients intolerant to statins or contraindicated) [75], a challenge arises from the reported meta-analyses, as the trials lasted a maximum of 52 weeks [76]. An open-label extension emerged to extend the drug evaluation period. This work indicates that the safety and efficacy results (a stable reduction of about 14% in LDL-C) persist for 2.5 years in patients with hypercholesterolemia and ASCVD and/or HeFH. This trial demonstrated that bempedoic acid effectively reduces LDL compared to a placebo, leading to improved global outcomes and long-term cardiovascular outcomes for patients with established ASCVD or those at high risk [76].

Regarding adverse reactions, bempedoic acid proves to be a safe drug. Notably, significant hepatic or muscular cytolysis occurs in only 2.8% compared with a placebo. Compared to statins, muscle damage is minimal because the enzyme it acts upon is predominantly concentrated in hepatocytes rather than muscle cells [75]. Severe complications, such as Achilles tendon rupture, have only been observed in patients with other associated risk factors [77]. The sole drug-related adverse effect is increased uric acid (mean rise is 0.7 mg/dL, 95% CI, 0.5–0.9 mg/dL) with a higher rate of gout flare (OR = 3.2; 95% CI, 0.12–8.2), primarily due to renal organic anion transporter 2 inhibition [78]. Therefore, evaluating uric acid before treatment and monitoring during treatment is recommended [79].

### 2.5. Pelacarsen

Pelacarsen is a new type of antisense oligonucleotide drug involved in reducing the level of Lp(a) by inhibiting the translation of mRNA of the Lp(a) gene in hepatocytes [80]. Lp(a), a lipoprotein similar to LDL in which ApoB is linked to Apo(a) [81,82], appears to be involved in the development of ASCVD. Although there is not yet a precise analysis of the role of Lp(a) in ASCVD due to a lack of FDA-approved pharmacological therapies, it should be considered an independent risk factor with values in the blood above 30 mg/dL to 50 mg/dL [83,84,85,86,87]. Lp(a) seems to exert this negative action in the development of atherosclerotic disease through three different ways: firstly, it carries out proinflammatory activity due to its high content of oxidized phospholipids; then, it has a prothrombotic effect due to the plasminogen-like protease domain on Apo(a), with a possible role also as an antifibrinolytic agent; finally, it has proatherogenic activity for the LDL-like moiety [88,89].

The first randomized trial to evaluate the use of a specific drug that acts on reducing Lp(a) levels was the IONIS-APO(a)Rx phase 2 trial. This work has shown that this new drug is able to reduce Lp(a) levels by between 67% and 72%; moreover, a simultaneous reduction was also noted in the overall levels of LDL-C, Apo(a), and ApoB, which finally leads to a reduction in the inflammatory activity of monocytes associated with oxidized phospholipids. The IONIS-APO[a]-LRX phase ½a trial showed how the conjugation of IONIS-APO(a)Rx with the GalNAc3 complex, mediating hepatocyte delivery via asialoglycoprotein [73], improves the potency of the drug by about 30 times, with an average reduction in Lp(a) blood levels of about 92.4%, also reducing the dose and without particular side effects. The AKCEA-APO(a)-LRx phase 2 trial, in which 286 patients with cardiovascular maladaptive disease and Lp(a) levels greater than 60 mg/dL were recruited, showed a dose-dependent reduction in patients treated with APO(a)-LRx compared to a placebo [88]. In particular, it has been seen how a dose of 20 mg per week is able to lead to an average reduction in Lp(a) blood levels of about 92% in the absence of serious adverse reactions such as flu-like syndrome, liver impairment, kidney damage, or thrombocytopenia. Finally, Lp(a) HORIZON, a phase 3 randomized controlled trial that started in 2019 and will probably end in 2024, aims to study the occurrence of major cardiovascular events as the primary outcome in a cohort of 8323 patients with blood levels of LPA greater than 70 mg/dl and randomly treated with pelacarsen or a placebo.

## 3. Novel Therapeutic Options That Are Not on the Market Yet

### 3.1. Olpasiran

Olpasiran is new si-RNA drug involved in the synthesis of Lp(a); in particular, it acts by inhibiting the translation of Lp(a) m-RNA. In preclinical studies in transgenic mice, olpasiran has been shown to lead to a dose-dependent decrease in serum Lp(a). In a phase 1 study, however, it showed an ability to reduce serum Lp(a) in a percentage between 72 and 97. The phase 2 OCEAN(a)-DOSE study demonstrated, in a cohort of patients randomly treated with 225 mg of olpasiran every 12 weeks or a placebo, a reduction in Lp(a) compared to low values of approximately 90% in the absence of significant adverse reactions [90].

### 3.2. ANGPTL3

Lipoprotein lipase (LPL) is a member of the lipase family situated on the luminal surface of capillaries that catabolizes plasma triglycerides of lipoproteins like chylomicrons and VLDL [91]. Its deficiency leads to severe hypertriglyceridemia [83,92] and a reduction in HDL-C [93], thereby increasing the risk of ischemic heart disease [94]. Angiopoietin-like proteins (ANGPTLs) are a family of proteins able to induce the posttranslational regulation of LPL, determining its inhibition [94]. Koishi R. [95] et al. investigated the role of angiopoietin-like protein 3 (ANGPTL3), a member of the vascular endothelial growth factor (VEGF) family expressed and secreted by the liver. Treating mice with this protein determined an increase in total cholesterol, triglycerides, and non-esterified fatty acids one day after injection, peaking after four days, suggesting an important potential role of ANGPTL3 in elevating serum lipids. This effect seems to be determined by an LDL-R-independent mechanism. Adam et al. [96] studied the inhibition process mediated by ANGPTL3 on LPL and endothelial lipase. A fully human monoclonal antibody, evinacumab, capable of inactivating ANGPTL3 in LDL-R-deficient mice, lowered the levels of triglyceride and cholesterol by derepressing LPL and EL, determining VLDL remodeling, resulting in the formation of lipid-depleted remnants, which were removed from circulation. This ended in a reduction in LDL-C in serum. EL, in the absence of LDL-R, seems to be the key to this process because the inhibition of this protein leads to an increase in LDL-C. In the ELIPSE HoFH [97] trial, the effect on LDL-C reduction in patients with homozygous familial hypercholesterolemia (HoFH) taking evinacumab versus a placebo was compared. At week 24, the patients in the evinacumab group registered a reduction of 47.1% in LDL-C, while in the placebo group, there was a 1.9% increase. The monoclonal antibody was well tolerated, and the most common adverse events were nasopharyngitis, influenza-like illness, dizziness, rhinorrhea, and nausea. In null–null homozygotes patients, there was a complete absence of LDL-R expression. Statins and PCSK9 inhibitors have a low-to-zero effect on cholesterol reduction, and evinacumab, approved by FDA for the treatment of HoFH, is already an important choice for them. Rosenson et al. [98], in a randomized double-blind phase 2 trial, examined 272 patients with refractory hypercholesterolemia, meaning patients with heterozygous familiar hypercholesterolemia or non-heterozygous familiar hypercholesterolemia with atherosclerotic cardiovascular disease; these patients were non-responders to other lipid-lowering therapies. The trial was designed in groups with different dosages of subcutaneous evinacumab (450 mg, 300 mg every week or every 2 weeks, or placebo) or intravenous evinacumab (15 mg/kg every 4 weeks, 5 mg/kg every 4 weeks, or placebo every 4 weeks). The reduction in LDL was observed from week 2 to week 16, and it was similar in the subcutaneous and the intravenous group, demonstrating the possibility of this human monoclonal antibody for lowering cholesterol not only in HoFH patients but also in every patient with refractory hypercholesterolemia. Other options in development for the inhibition of ANGPTL3 are targeting its RNA. Vupanorsen (AKCEA-ANGPTL3-L) is an antisense oligonucleotide that targets ANGPTL3 mRNA [99]. It showed promising results in lowering triglycerides and LDL, but recently, considering the insufficient results and adverse reactions, the development program was discontinued [100]. Another example is the investigational RNAi ARO-ANG3 trial, which has now completed its second phase; the key findings were a mean reduction in ANGPTL3 of up to 82%, LDL-C of up to 48.1%, and ApoB of up to 39.2% in patients with HoFH [101]. No serious adverse reactions were identified.

### 3.3. CETP

Cholesteryl ester transfer protein (CETP) is a hydrophobic glycoprotein produced in the liver and adipose tissue that catalyzes the transfer of cholesteryl esters from HDL to triglyceride-rich lipoproteins (LDL, VLDL) [102]. Patients with a CETP deficiency have increased HDL-C [103], which has a beneficial cardiovascular role, as HDL particles are considered to have anti-inflammatory, anti-oxidative, anti-apoptotic, and anti-thrombotic roles, along with a reduction in non-HDL cholesterol [104,105]. This evidence has led to an increasing interest in the inhibition of CETP to reduce the impact of cholesterol on the cardiovascular system; following this idea, several drugs have been developed.

Hovingh G.K. [106] et al. conducted a randomized, double-blind phase 2 study on obitracepib (TA-8995), a CETP inhibitor, with the conclusion that 10 mg of this drug alone reduced LDL-C by up to 45.3% and by up to 68% in patients receiving obitracepib plus 20 mg of atorvastatin; HDL-C increased by up to 179% in the 10 mg obitracepib group. A similar result was achieved in another randomized, double-blind, placebo-controlled trial by Nicholls et al. [107]; in a group of 120 patients with a background of high-intensity statin medication, the add-on treatment with 10 mg of obitracepib significantly reduced LDL, Lp(a), and ApoB and increased HDL-C. The most common adverse reactions were nausea and headache.

The effect of this drug seems very promising for reducing cardiovascular deaths; several studies are evaluating this subject: PREVAIL [108] is a placebo-controlled, double-blind, randomized phase 3 study consisting of 9000 patients that has been focusing on the effect of 10 mg of obitracepib on patients with a history of ASCVD not controlled by traditional lipid-lowering therapies; REVEAL [109] is another randomized phase 3 trial with almost 30,000 patients with previous cardiovascular disease already taking a statin; they also received placebo or another CETP inhibitor, anacetrapib. The study has already demonstrated a significant reduction in the primary endpoint of coronary death and myocardial infarction or coronary revascularization. No severe side effects were highlighted [110]. At the conclusion of these two trials, we will have novel information on the potential use of these medications. CETP inhibitors could represent another choice of therapy for several patients with a high cardiovascular risk who are intolerant to statins or that may need a more impactful reduction in cholesterol not achieved with other therapies.

Figure 2 illustrates a visual representation of the mechanisms of the drugs listed above.

### 3.4. HMG-CoA Reductase Degrader

The introduction of statins to the market has changed cholesterol-lowering therapies; these drugs are still the most commonly prescribed thanks to their low cost and great benefits. Statins bind to and inhibit the catalytic domain of their target, the 3-hydroxy-3-methylglutaryl–coenzyme A reductase (HMGCR) [111], stopping the conversion of HMG-CoA to mevalonate, a key reaction for cholesterol synthesis. Mevalonate derivatives are known to control HMGCR levels through a multivalent feedback mechanism, and in the absence of serum cholesterol, there is a higher activity of HMGCR [112]. Statins, by blocking this reductase, induce a compensatory increase in HMGCR [113] that can paradoxically reduce the effect of the drug in lowering cholesterol. Jiang et al. [114] identified a specific HMGCR degrader, Cmpd81; LDL mice treated with Cmdp81 alone or lovastatin and Cmpd81 showed a decrease in VLDL and LDL in their liver cells and fewer atherosclerotic plaques. No effects on body weight, food intake, nor cardiovascular toxicity were associated with Cmdp81 during the study. This is a promising new approach for reducing cholesterol alone or in a synergistic way in patients already in treatment with statins.

### 3.5. ASGR1

The asialoglycoprotein receptor (ASGPR) was discovered in the mid-1960s by Ashwell, Morell, and colleagues [115,116] in their studies of the uptake of serum glycoproteins by hepatocytes. It is mainly expressed in the liver, where it is responsible for the endocytosis and degradation of desialylated glycoproteins with terminal galactose (Gal) or N-acetylgalactosamine (GalNAc) residues, making its most interesting role the removal of desialylated forms of glycoproteins from circulation [117]. ASGPR consists of two subunits: ASGR1 and ASGR2 [118]; the first and major one has been investigated for its potential role in regulating serum cholesterol.

Nioi et al. [119] discovered that two loss-of-function variants in ASGR1, del12 and p.W158X, are associated with a non-HDL-C reduction, an increase in levels of alkaline phosphatase and vitamin B12, and a reduced risk of coronary artery diseases. Wang et al. [120] revealed that the inhibition of ASGR1 in mice determined the impossibility to internalize asialoglycoproteins, activating the enzyme AMPK, and finally increasing LXRα, upregulating the ATP-binding cassette transporter (ABCA1), which delivers cholesterol to HDL, and ABCG5/G8, which expels cholesterol to the bile; AMPK also inhibits the transcription factor SREBP and consequently blocks lipogenesis. ASGR1 mice liver cells displayed lower triglycerides and serum cholesterol compared to wild-type mice and an increase in biliary cholesterol in the gallbladder, posing a potential risk for gallstones.

Inhibition of ASGR1 could represent an interesting future approach to reducing cardiovascular risk by lowering cholesterol through its excretion into the biliary system; the localization of the receptor is mainly in the liver, which could avoid adverse reactions depending on AMPK activation in other organs. Additionally, another advantage is that ASGR1 inhibitors synergize with other cholesterol-lowering drugs like statins and ezetimibe, providing other choices for therapy. The limit of this potential drug is its role in balancing the uptake of glycoproteins that may cause hepatic damage [121], cholelithiasis, and the interaction with platelet function [122].

The first randomized, double-blind, phase 1 trial with 48 participants was conducted in 2017 using a specific ASGR1 inhibitor, the human monoclonal antibody AMG-529 [123,124], to assess the potential risk of this treatment. The antibody showed an acceptable safety profile, and the most common reaction was a headache, with an increase in alkaline phosphatase noted. More studies in the future will clarify the benefits and disadvantages of this ASGR1 inhibition.

### 3.6. PCSK9 Vaccine

PCSK9 inhibition is considered, since the approval of evolocumab, to be one of the most impactful therapies regarding cholesterol reduction, but due to the elevated cost and frequent administration, in recent years, another idea that has been explored is developing a vaccine capable of inducing an immune response against PCSK9 and therefore blocking its function. Pan et al. [125] designed and injected a PCSK9 vaccine in mice; five PCSK9 epitope peptides were conjugated with Qβ-virus-like-particles (VLP), creating the PCSK9Qβ-003 vaccine. Administering this medication to mice determined a decrease in triglycerides and non-HDL cholesterol after the third dose; the vaccine was able to inhibit LDLR degradation mediated by PCSK9, it synergized with statins, and no significant adverse reactions were found in the mice treated with the vaccine. Another recent randomized, double-blind, placebo-controlled phase 1 study [126,127] is evaluating the effect of VXX-401, a vaccine designed to stimulate the immune response against PCSK9 in humans; it has already been tested in rats, pigs, and monkeys with sufficient results. The 48 participants, naïve to statins and divided into four treatment cohorts, have been receiving a dosage of 100 mcg or 300 mcg of the drug and two placebo groups. At the conclusion of this trial, we will have more information on the effect and safety of this promising PCSK9 inhibitor in humans.

### 3.7. Additional Emerging Strategies

An oral PCSK9 inhibitor, MK-0616, was the subject of a phase 2, randomized, double-blind, placebo-controlled study [128,129]. The promising drug has the same mechanism as the other PCSK9 inhibitors with the advantage of being administered once a day orally. A total of 381 participants with hypercholesterolemia were divided into four groups receiving a different dosage of MK-0616, and after 8 weeks, they showed a significant reduction in LDL-C (60.9% in the 30 mg group) compared to a placebo; no serious adverse reactions were reported.

Clustered regularly interspaced short palindromic repeats is a DNA base editor that functions without double-strand breaks and determines A->G edits in target DNA; this technology was used to design a specific PCSK9 editor. With a single infusion, it knocked down PCSK9 in almost every hepatocyte in cynomolgus monkeys and lowered LDL-C by up to 60% for a period of 8 months. Similar results were obtained in another study with non-human primates [130]. These promising results have determined the start of a phase 1 single-ascending dose to evaluate the safety of VERVE-101, a CRISPR base-editing drug assembled by a messenger RNA for an adenine base editor and a guide RNA and designed to block PCSK9 in patients with HeFH, atherosclerotic cardiovascular disease, and uncontrolled hypercholesterolemia [131].

A summary of the available and emerging LDL-C-lowering therapies is provided in Table 1.

## 4. Conclusions

ASCVD stands out as the most impactful cause of morbidity and mortality globally, particularly among the elderly [132]. In 2019 alone, an estimated 17.9 million deaths were attributed to CVDs, representing 32% of all causes of death during that year [132]. Mitigating ASCVD remains a formidable challenge in modern medicine, which is tightly linked to behavioral risk factors such as tobacco use, an imbalanced and unhealthy diet, and alcohol abuse, exacerbated by insufficient physical activity. While intervening in these factors is crucial, it often proves inadequate in shielding individuals from heart attacks and strokes, the predominant causes of CV death [133,134,135].

The advent of statins over 30 years ago marked a pivotal breakthrough in LDL-C-lowering drugs, with atorvastatin remaining one of the most widely prescribed medications globally. Unfortunately, its efficacy falls short in individuals intolerant to statins or those failing to achieve LDL targets through statin monotherapy. Over the past two decades, a slew of novel medications aimed at combatting ASCVD have secured approval and entered the market. Noteworthy among them are ezetimibe, targeting intestinal cholesterol absorption and often used in combination with statins; mipomersen and lomitapide, prescribed to reduce LDL-C in patients with HoFH; and evolocumab and alirocumab, potent PCSK9 inhibitors exerting a significant impact on cholesterol levels, which is integral in secondary therapies when statins prove insufficient. Recent additions to this roster include bempedoic acid, which inhibits LDL synthesis in the liver; inclisiran, another formidable PCSK9 inhibitor; and evinacumab, which disrupts ANGPTL3, impeding its ability to decelerate fat-degrading enzymes.

Future prospects in cholesterol-lowering therapies hold promise with the emergence of obicetrapib, a CETP inhibitor, which concurrently lowers LDL-C and elevates HDL-C, either alone or in combination with statins; ARO-ANG3, an RNAi agent akin to evinacumab, which inhibits ANGPTL3; and olpasiran, a small-interfering RNA, and pelacarsen, an antisense oligonucleotide, which prevent the assembly of Lp(a), a recognized cardiovascular risk factor [136] historically unaddressed by specific cholesterol-reducing therapies. Drugs such as PCSK9 inhibitors, mipomersen, and CETP inhibitors are known to lower Lp(a).

Exploring additional avenues, potential medications in development include ASGR1 inhibitors, which enhance cholesterol efflux to bile and consequently reduce serum levels, and HMGCR degraders that optimize statin effects by mitigating compensatory increases in HMGCR. This diverse array of emerging therapies holds the potential to substantially lower serum cholesterol levels, alleviating the burden of cardiovascular diseases.

## Figures and Tables

**Figure 1 biomedicines-12-00432-f001:**
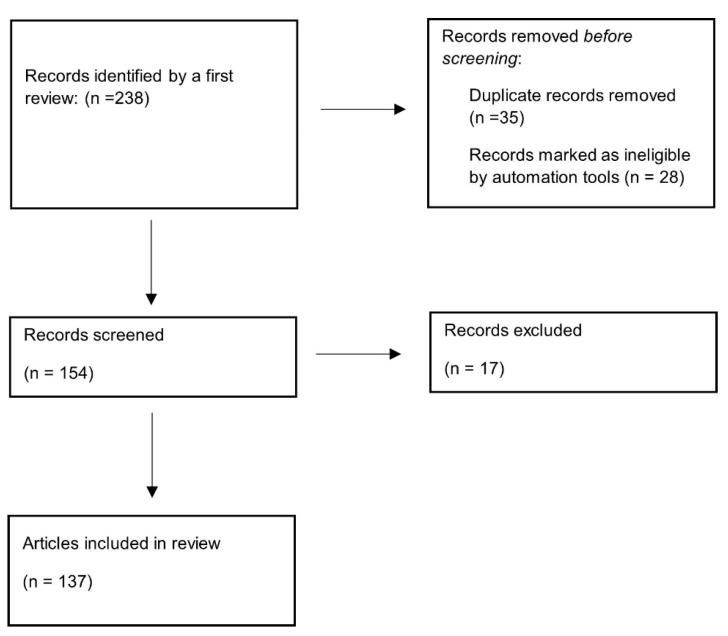
Flow chart.

**Figure 2 biomedicines-12-00432-f002:**
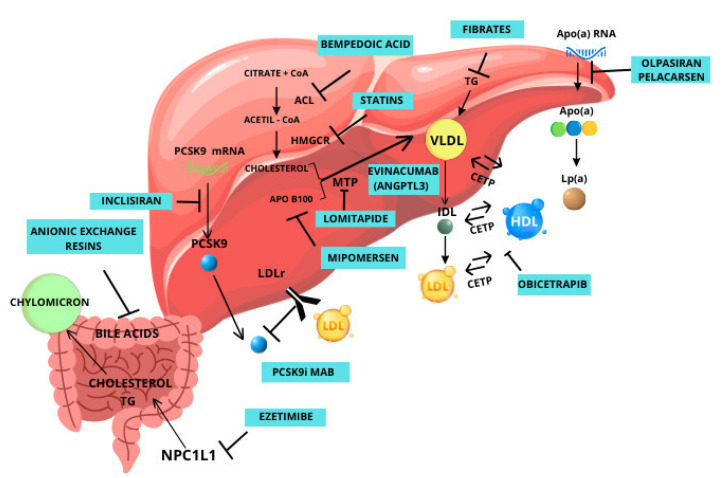
The mechanisms of the main cholesterol-lowering drugs.

**Table 1 biomedicines-12-00432-t001:** Available and emerging LDL-C-lowering therapies.

Name	Target	Phase	LDL-C Reduction	Adverse Reactions	Mechanism	Clinical Use or Perspectives
**Statins** [12,13,14,15,16]	HMGCR	Approved	20% to 50%	Rhabdomyolysis	Competitively inhibits HMG-CoA reductase	Primary and secondary prevention of ASCVD Severe hypercholesterolemia
**Ezetimibe** [20,21,22]	NPC1L1	Approved	23%	Stomach pain, mucle pain and cramps, asthenia	Inhibits the intestinal absorption of cholesterol by blocking NPC1L1	Add-on to statin therapy or alone for ASCVD and for severe hypercholesterolemia
**Mipomersen** [35,36,37,38]	ApoB100 mRNA	Approved, FDA only	26%	Injection-site reactions, flu-like symptoms, ALT elevations	Anti-sense oligonucleotide that prevents the production of apolipoprotein B	HoFH treatment
**Lomitapide** [41,42,43,44,45,46,47,48,49,50]	MTP	Approved	40% to 50%	Liver toxicity, GI adverse reactions	Blocks MTP protein	HoFH treatment
**PCSK9i antibody** [27,28,29]	PCSK9	Approved	47%	Nasopharyngitis, upper respiratory tract infection, back pain, joint pain, flu-like symptoms, injection-site reactions	Inhibits PCSK9, preventing its interaction with cholesterol receptors	HoFH treatment and prevention of ASCVD in association with statins or alone.
**Bempedoic acid** [62,63,64,65,66,67,68,69,70,71,72,73,74,75,76,77,78,79]	ACL	Approved	17% to 21%	Hyperuricaemia, pain in arms or legs, anaemia	Blocks adenosine triphosphate citrate lyase	Add-on to statin for HeFH and prevention of ASCVD
**Inclisiran** [51,52,53,54,55,56,57,58,59,60,61]	PCSK9 mRNA	Approved	50%	Pain and rash at the injection site.	Reduces the production of PCSK9 through gene silencing	Add-on to statin for HeFH and prevention of ASCVD
**Evinacumab** [101,102]	ANGPTL3	Approved	47%	Nasopharyngitis, influenza-like illness, dizziness, rhinorrhea, nausea.	Stops ANGPTL3 from blocking vascular lipases that break down fats	HoFH treatment
**Pelacarsen** [80,81,82,83,84,85,86,87,88]	LPA mRNA	Phase 3	26%	Flu-like syndrome, liver impairment, kidney damage, thrombocytopenia	Blocks translation of mRNA of the LPA gene	Prevention of ASCVD
**Olpasiran** [90]	Lp(a)	Phase 3	Lp(a) < 90%	Injection-site pain	Small-interfering RNA that prevents the assembly of Lp(a)	Reduction in Lp(a)
**Obicetrapib** [110]	CETP	Phase 3	45%	Nausea, urinary tract infection, headache	Inhibits CETP, which catalyzes the transfer of cholesteryl esters from HDL to LDL and VLDL	Reduction in LDL and apoB and increase in HDL
**HMARO-ANG3** [99]	ANGPTL3 mRNA	Phase 2	44–48%	Headache, respiratory tract infections, local injection-site reactions.	RNAi, which inhibits ANGPTL3	Treatment of dyslipidemias, familiar hypercholesterolemia, and hypertriglyceridemia.
**MK-0616** [128,129]	PCSK9	Phase 2	60.9%	Arthralgia, diarrhea, nausea, dyspepsia	Oral PCSK9 inhibitor	Treatment of hypercholesterolemia
**PCSK9 vaccine (VXX-401)** [126,127]	PCSK9	Phase 1	30–50% *	No damage detected *	Induces immune response against PCSK9, blocking it	Treatment of hypercholesterolemia by inducing antibodies against PCSK9
**VERVE-101** [131]	PCSK9 gene	Phase 1	69% *	Elevations in liver function tests *	Inhibits PCSK9 through a CRISPR base-editing technique	Treatment of HeFH, hypercholesterolemia, and ASCVD.
**ASGR1i** [121,122,123,124]	ASGR1	One phase 1 study in 2017		Potential liver toxicity	Increases cholesterol efflux to bile	Treatment of hypercholesterolemia and ASCVD.
**HMGCR degrader** [114]	HMGCR	No human trial		No damage detected°	Reduces statin-induced HMGCR accumulation	Reduction in cholesterol

ACL: ATP citrate lyase; ANGPTL3: angiopoetin-like 3 protein; CETP: cholesteryl ester transfer protein; FDA: Food and Drug Administration; ASCVD: atherosclerotic cardiovascular disease; HMGCR: 3-hydroxy-3-methylglutaryl–coenzyme A reductase; HoFH: homozygous familial hypercholesterolemia; HeFH: heterozygous familiar hypercholesterolemia; VLDL: very-low-density lipoprotein; LDL-C: low-density lipoprotein; HDL: high-density lipoprotein; LP(a): lipoprotein (a); MTP: microsomal triglyceride transfer protein; NPC1L1: Niemann–Pick-like protein 1C1; PCSK9i: proprotein convertase subtilisin kexin type 9 inhibiting; ASGR1i: asialoglycoprotein receptor inhibitor; ALT: alanine aminotransferase; GI: gastrointestinal; CRISPR: clustered regularly interspaced short palindromic.

## Data Availability

No dataset was generated for the publication of this article.

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
