# Peer review of "Advances in Pharmacological Approaches for Managing Hypercholesterolemia: A Comprehensive Overview of Novel Treatments"

_biomedicines, 2024, doi:10.3390/biomedicines12020432_

Round 1

Reviewer 1 Report

Comments and Suggestions for Authors

Overall/General Comments

In the present study, the authors provide a thorough exploration of the recent advancements in hypercholesterolemia management​​. Overall, an interesting and well written review focusing on novel therapeutic agents, beyond traditional statins, for managing hypercholesterolemia.

Specific Comments

1. The authors should consider adding commentary on the availability and extent of studies for each drug discussed regarding their effectiveness in reducing atherosclerotic cardiovascular disease (ASCVD) events. This addition would provide a more comprehensive understanding of the clinical relevance of these treatments in the context of ASCVD risk reduction.

2. If feasible, the inclusion of a figure summarizing the mechanisms of action of each drug discussed in the paper could greatly enhance the reader's understanding. Such a visual representation would provide a quick reference for the location and mode of action of each treatment within the cholesterol metabolism pathway. This addition would be particularly useful for visually oriented readers and could aid in better illustrating the comparative and complementary roles of these novel therapies.

Author Response

Overall/General Comments

In the present study, the authors provide a thorough exploration of the recent advancements in hypercholesterolemia management​​. Overall, an interesting and well written review focusing on novel therapeutic agents, beyond traditional statins, for managing hypercholesterolemia.

Specific Comments

1.The authors should consider adding commentary on the availability and extent of studies for each drug discussed regarding their effectiveness in reducing atherosclerotic cardiovascular disease (ASCVD) events. This addition would provide a more comprehensive understanding of the clinical relevance of these treatments in the context of ASCVD risk reduction.

Re: Thank you for your feedback. We have addressed the issue by incorporating comments regarding the drug's effects on reducing ASCVD events wherever this information was previously lacking. These specific characteristics have been drawn from the most recent studies on the topic, as referenced in the text.

  1. If feasible, the inclusion of a figure summarizing the mechanisms of action of each drug discussed in the paper could greatly enhance the reader's understanding. Such a visual representation would provide a quick reference for the location and mode of action of each treatment within the cholesterol metabolism pathway. This addition would be particularly useful for visually oriented readers and could aid in better illustrating the comparative and complementary roles of these novel therapies.

Re: Thank you for your valuable suggestion. We concur on the importance of a visual representation of the concepts in the manuscript, and accordingly, we have created a figure that fulfills these requirements. We trust that it will meet your expectations.

Reviewer 2 Report

Comments and Suggestions for Authors

The current manuscript is interesting, however, there are a some of points to be improved.
Major comments:
1.    A flowchart of the initial publications (PubMed, MEDLINE) search is needed, how many papers from this search are included in the review? How many papers were omitted? Was the search done by more than 1 person?
2.    Mention of drug brend names is generally not allowed in reviews and scientific articles; It is allowed to use only medicinal compounds corresponding to International nonproprietary names (page 5:  NILEMDOR, NUSTENDIR).

Minor comments:
1.    Table 1 is not mentioned in the text.

Author Response

Reviewer 2

The current manuscript is interesting, however, there are a some of points to be improved.
Major comments:
1.    A flowchart of the initial publications (PubMed, MEDLINE) search is needed, how many papers from this search are included in the review? How many papers were omitted? Was the search done by more than 1 person?

RE: Thank you for your comment. We have addressed it by incorporating a flow chart in the text that describes the method of data collection.

  1. Mention of drug brend names is generally not allowed in reviews and scientific articles; It is allowed to use only medicinal compounds corresponding to International nonproprietary names (page 5:  NILEMDOR, NUSTENDIR).

RE: Thank you for bringing this to our attention. We have addressed the concern by removing the brand name of the drug.

Minor comments:
1.    Table 1 is not mentioned in the text.

RE: I apologize for the oversight. The table has been included in the new version of the manuscript.

Reviewer 3 Report

Comments and Suggestions for Authors

The manuscript entitled “Advances in Pharmacological Approaches for Managing Hypercholesterolemia: A Comprehensive Overview of Novel Treatments” reviewed that currently available drugs for the treatment of hypercholesterolemia. However, the content of the article is mainly a description and listing of various drugs, which lacks emphasis and is not interesting and meaningful enough.

Author Response

Reviewer 3

The manuscript entitled “Advances in Pharmacological Approaches for Managing Hypercholesterolemia: A Comprehensive Overview of Novel Treatments” reviewed that currently available drugs for the treatment of hypercholesterolemia. However, the content of the article is mainly a description and listing of various drugs, which lacks emphasis and is not interesting and meaningful enough.

RE: Thank you for taking the time to review our manuscript. We acknowledge your concern regarding the emphasis and meaningfulness of the content. We understand the importance of providing valuable insights and perspectives to the readers. In response to your feedback, we have revised the manuscript to ensure a more engaging and insightful discussion, with a focus on the practical implications and potential advancements in the field. We appreciate your input and we have enhanced the overall quality of the manuscript. Your valuable suggestions have undoubtedly contributed to the improvement of our work.

Reviewer 4 Report

Comments and Suggestions for Authors

Mormone and coworkers summarized the early and the latest issues about the LDL-C reducing drugs which partially are now commercially available and partially are standing in front of the market. Alternative therapeutic targets are also discussed.

Specific comments:

1. I miss a clear and detailed aim at the end of the introduction. In addition, possibly mention of the authors publication with ‘our research group investigated’, ‘our findings support that’ etc. may help to create a more personal attitude. Dozens of papers are published per year in this field, these phrases may help to draw the attention of the readers.

2. Creating attractive coloured figures about the less-known mechanisms (bempedoic acid, ANGPTL3 inhibitors, ASGR1 inhibitors) may also point out the importance of the manuscript.

3. Mention of other subheads is considered. (e.g. Currently available drugs vs. Novel therapeutic options which are not on the market yet)

General:

1. Some typos and failures were observed which I labelled with green in the text, please see in the enclosed file.

2. Page 5. Use of brand names are not recommended.

3. Table 1 is not labelled in the text and needs some improvement. For example, references should include to the table; and all abbreviation should include to the footnote (ALT, GI, CRISPR, ASGR1i…). Since this is a table which prominently mention data of LDL-C reduction, olparisan should remove from the table (Lp(a) reduction) OR if olparisan remains, pelacarsen would also mention.

4. Ref 76,90,94,130 are lacking in the reference list. The manuscript contains 141 references which showed some discrepancies from ref88. I feel that the references were probably entered to the text by hand; or the reference list was not updated before submitting. Please, check the references once again very carefully and the use of a reference manager is highly recommended.

Comments on the Quality of English Language

Minor editing of English language required, please, see attachment.

Author Response

Reviewer 4

Mormone and coworkers summarized the early and the latest issues about the LDL-C reducing drugs which partially are now commercially available and partially are standing in front of the market. Alternative therapeutic targets are also discussed.

Specific comments:

1.I miss a clear and detailed aim at the end of the introduction. In addition, possibly mention of the authors publication with ‘our research group investigated’, ‘our findings support that’ etc. may help to create a more personal attitude. Dozens of papers are published per year in this field, these phrases may help to draw the attention of the readers.

Re: Thank you for your comment. We have addressed it by adding the aim of the study at the end of the introduction. In addition, we have emphasized our results accordingly.

  1. Creating attractive coloured figures about the less-known mechanisms (bempedoic acid, ANGPTL3 inhibitors, ASGR1 inhibitors) may also point out the importance of the manuscript.

Re: Thank you for this important suggestion. We have created Figure 2, which illustrates the mechanisms of the drugs. We trust that it will meet your expectations.

  1. Mention of other subheads is considered. (e.g. Currently available drugs vs. Novel therapeutic options which are not on the market yet)

Re: Thank you for your suggestion. We have incorporated the subhead in the new version of the manuscript.

General:

  1. Some typos and failures were observed which I labelled with green in the text, please see in the enclosed file.

Re: Thank you for your revision. We have addressed and corrected all the mistakes as per your alerts in the text.

  1. Page 5. Use of brand names are not recommended.

RE: Thank you for your comment. We have addressed it by removing the brand name of the drug from the text.

  1. Table 1 is not labelled in the text and needs some improvement. For example, references should include to the table; and all abbreviation should include to the footnote (ALT, GI, CRISPR, ASGR1i…). Since this is a table which prominently mention data of LDL-C reduction, olparisan should remove from the table (Lp(a) reduction) OR if olparisan remains, pelacarsen would also mention.

RE: Thank you for your comment. We have enhanced Table 1 in accordance with these instructions.

  1. Ref 76,90,94,130 are lacking in the reference list. The manuscript contains 141 references which showed some discrepancies from ref88. I feel that the references were probably entered to the text by hand; or the reference list was not updated before submitting. Please, check the references once again very carefully and the use of a reference manager is highly recommended.

RE: We apologize for this oversight. We have now reviewed all references and corrected any discrepancies.

Round 2

Reviewer 2 Report

Comments and Suggestions for Authors

The authors supplemented the manuscript with the necessary explanations. The Figure 2 improved the readability of the manuscript.
The manuscript may be accepted in its present form.

Reviewer 3 Report

Comments and Suggestions for Authors

Dear editor, I checked the current PDF version, and I do not have other concerns about this paper, THX

Reviewer 4 Report

Comments and Suggestions for Authors

The manuscript has been improved based on the referees' suggestion and it suggests for publication.